# Measurement Method for Evaluating the Lockdown Policies during the COVID-19 Pandemic

**DOI:** 10.3390/ijerph17155574

**Published:** 2020-08-02

**Authors:** Mohammed Al Zobbi, Belal Alsinglawi, Omar Mubin, Fady Alnajjar

**Affiliations:** 1School of Computer, Data and Mathematical Sciences, Western Sydney University, Parramatta South Campus, Sydney 2116 NSW, Australia; M.AlZobbi@westernsydney.edu.au (M.A.Z.); b.alsinglawi@westernsydney.edu.au (B.A.); O.Mubin@westernsydney.edu.au (O.M.); 2College of Information Technology, United Arab Emirates University, Al Ain P.O. Box 15551, UAE

**Keywords:** COVID-19, infectious disease modeling, basic reproduction number, machine learning, government regulations, spread control

## Abstract

Coronavirus Disease 2019 (COVID-19) has affected day to day life and slowed down the global economy. Most countries are enforcing strict quarantine to control the havoc of this highly contagious disease. Since the outbreak of COVID-19, many data analyses have been done to provide close support to decision-makers. We propose a method comprising data analytics and machine learning classification for evaluating the effectiveness of lockdown regulations. Lockdown regulations should be reviewed on a regular basis by governments, to enable reasonable control over the outbreak. The model aims to measure the efficiency of lockdown procedures for various countries. The model shows a direct correlation between lockdown procedures and the infection rate. Lockdown efficiency is measured by finding a correlation coefficient between lockdown attributes and the infection rate. The lockdown attributes include retail and recreation, grocery and pharmacy, parks, transit stations, workplaces, residential, and schools. Our results show that combining all the independent attributes in our study resulted in a higher correlation (0.68) to the dependent value Interquartile 3 (Q3). Mean Absolute Error (MAE) was found to be the least value when combining all attributes.

## 1. Introduction

Three major global pandemic outbreaks have spread in the last few decades, severe acute respiratory syndrome (SARS), Middle East respiratory syndrome (MERS), and Ebola virus (EVD). SARS and MERS are both caused by coronaviruses. This group of crown-like viruses can cause liver, neurologic, gastrointestinal, and respiratory disease [1,2]. SARS first appeared in Southern China in November 2002, caused by the SARS-associated Coronavirus (SARS-CoV) with ready transmission through droplets. The incubation period was typically 2–7 days. The outbreak was declared to be over by late July 2003; this global pandemic resulted in a total of 8098 probable cases with 774 associated deaths [3]. Compared to SARS, new MERS cases were being reported. Since it was first identified in Saudi Arabia in 2012, a total of 2519 laboratory-confirmed MERS cases including 866 associated deaths, were reported globally by the end of January 2020 [4]. With 79.6% similarity in sequence identity as SARS-CoV, the newest coronavirus (COVID-19) outbreak first started in Wuhan, China on 12 December 2019 [5,6]. Considering the rapid spread over the globe and the person-to-person transmission capability, the World Health Organization (WHO) declared COVID-19 to be a Public Health Emergency of International Concern on 1 February 2020 [7]. As lockdown restrictions continue to ease in countries worldwide, there is a need to avoid another wave of the outbreak that may occur in any place and at any time [8]. The second wave may not be as large as the first one, but it may lead to another worldwide outbreak, especially when reducing restrictions on air travel. For this reason, governments need to be more vigilant and should closely monitor any possibility of an outbreak. Predicting the spread of disease globally can foresee its impact, effectively manage the supply chain prior to the epidemic, and secure allocations for critically affected and at-risk countries, thus reducing the shortage issue for endangered health workers on the frontline [9].

The aim of this paper is to highlight some lockdown factors that may have a direct impact on the contagion level. The infection and mortality rates not only depend on the contagion level but also many other factors such as a late response at the start of the pandemic. Lockdowns have been applied by most governments worldwide, either as curfews or on a voluntary basis. The community lockdown can be statistically defined as maintaining the stay at home duration for as long as possible during 24 h.

## 2. Background

There are similarities between SARS, MERS, and EVD. However, none of the mentioned diseases have caused a global outbreak as COVID-19 has. It is hard to predict the major circumstances that gave COVID-19 a good chance to spread that fast. Table 1 explains the differences between these diseases and partially points out major spreading reasons. 

### Possible Pandemic Causes

A quick comparison between the four viruses may conclude that MERS and COVID-19 are very similar. COVID-19 has the highest recorded infection rate. The basic reproduction number measures the infections produced by an infected individual in a population that is totally susceptible, and it is known as Pronounced R-naught (R0). This number is important in determining how contagious a disease is, or how quickly an infection will spread through a population [10].

The reason for COVID-19 spread in a large cluster may refer to other factors that are not confirmed yet. It is obvious that there is a minimum number of infections required to ignite any outbreak, this is large enough in urban areas and crowded cities. In a rough estimation, it is enough to infect a minimum of 0.2% of the inhabitants of many cities to reach the outbreak threshold value. This estimation depends on the R0 value of the virus and the population density. For instance, 0.2% is not enough to create an outbreak in suburban or fewer density cities. Wuhan/Hubei has one of the highest increases in population rate in China; its average annual increase rate is 7.9% with a population of 11 million [11]. The density of the city center of Wuhan is unknown, but the average of the world's medium metropolitans is around 29,000 people per kilometer. This number is usually doubled or tripled during the daytime in the city center.

Various factors created an ideal environment for the virus to spread: the city center density, the outbreak core center, the high seasonal shopping celebrating the upcoming Chinese spring festival, the large number of international students, and the worldwide visitors. In contrast, MERS and SARS most probably started in suburban areas [12]. MERS has appeared in places far away from main cities in Saudi Arabia since camel farms are available in unoccupied areas of deserts.

## 3. Materials and Methods

This article presents some suggested methods for measuring the R0 spread in the short run. This is to provide better measurement tools to government bodies, so they can promptly decide in accordance with the spread development. During the past few months, the spread measurements have adopted the value of R0 = 2.5 [13]. This value was calculated during the initial stages of the pandemic. The calculation method takes into consideration an incubation period of 14 days for the virus. As time passed, the R0 rate dramatically dropped down due to lockdown procedures, self-quarantine, and people’s awareness of the disease. Moreover, the uncertainty in understanding the infection numbers, in the early stages was high. Currently, the uncertainty percentage rate is lower, and the scan/testing rate of suspected patients is higher. Also, the virus test result period is much shorter than before, it may not exceed a few hours. We need to find a faster and more accurate measurement tool that gives governments a better understanding of the outbreak growth. The measurement model in this paper comprises the measurement of lockdown procedures, which may support government decisions. The method adopts a short period of measurement, which should not exceed four days. This may not seem accurate for measuring R0, but it can be considered another proposed model of measurement. This is essential to monitor the social distancing efficiency closely. Four days may allow the prompt review of some public health actions for better spatial protection against this disease.

Governments should have a long-term plan for this outbreak; this includes continuous revision of the lockdown policies, restrictions, and procedures. The policies should compromise between preventing an over-limit outbreak and mitigating economic loss. Limited capacity is defined as the ultimate capacity for intensive care units in the public health system. Governments are concerned about finding the optimal planning response since they are unable to determine the outbreak spread time and the social activities that cause the highest spread. Future modeling will account for the actions taken by governments, which include restricting travel, isolating people with the virus and their contacts, social distancing, growing health system capacity, and others [14]. For approximate evaluation, we built a naïve model that may support decision making. The model should provide a short period of reading the exponential spread of the disease. A period of four days is defined as a short time to re-review the public health policy. The governments should therefore respond promptly according to the resulting output.

### Measuring R0

To implement a proper model for measuring the lockdown efficiency, we introduced a model to measure some factors that have a direct impact on lockdown procedures. The R0 value was calculated daily, and then it was grouped by interquartile ranges. The R0 can be calculated by subtracting the number of infections in the previous day from the current day infections. The growth rate is given by $\frac{x_{i}-x_{\left(i-1\right)}}{x_{\left(i-1\right)}}$. Suppose that the number of infections is denoted by *X*, where *X* is defined as:(1)$$X=\text{\{}x_{0},x_{1},x_{2},\ldots .x_{n}\}$$
where $x_{n}$ represents the number of infections on the *n*th day.

The dispersion value is calculated by finding the interquartile mean of R0 for each interquartile range. Two interquartile values of Q1 and Q3 are found as follows:(2)Q1=N/4
(3)Q3=3×Q1
where *N* denotes the number of days.

Three interquartile ranges are defined by finding the mean of the values between zero and Q1, then the mean of the values between Q1 and *Q*3, and finally the mean of all the values greater than Q3. Close monitoring of the mean values of R0 would give a good indication of the daily social distancing efficiency. The mean value is not unique, in this case, it is a value that keeps changing daily. Calculating the mean value on a daily basis is essential to reduce the error rate caused by underestimating the actual number of daily infected cases. The mean value for each quartile is defined as:(4)$${\bar{x}}_{Qp}=\{\begin{array}{c} \frac{\sum_{i=1}^{Q1}\;\frac{x_{i}-\;x_{\left(i-1\right)}}{x_{\left(i-1\right)}}}{Q_{1}+1},\;\;\;\;p=1\\ \frac{\begin{array}{c} \sum_{i=Q1+1}^{Q3}\;\frac{{\dot{x}}_{i}-\;{\dot{x}}_{\left(i-1\right)}}{x_{\left(i-1\right)}} \end{array}}{Q_{3}-Q_{1}},\;\;\;p=2\\ \frac{\begin{array}{c} \sum_{i=Q3+1}^{N}\;\frac{{\dot{x}}_{i}-\;{\dot{x}}_{\left(i-1\right)}}{x_{\left(i-1\right)}} \end{array}}{N-Q_{3}},\;\;\;p=3 \end{array}$$
where $\bar{x}$ donates the mean values for each interquartile, and p represents the quarter number.

## 4. Results and Discussion

Finding out methods for gauging the contagion rate R0 can be a complicated task. Also, finding out methods for measuring the major factors that have a direct impact on R0 is an even harder task. We introduced a methodology for evaluating R0 within a short period by calculating the dispersion value of R0 interquartile. This may provide better accuracy for the latest R0 value. The second part of the methodology was finding the direct factors that may affect the R0 value. For this purpose, Google community mobility reports were used.

Google reports for social distancing are used in this measurement tool. Google has developed a COVID-19 Community Mobility Report for various countries. They abstract data from the length of stay at different locations. The length of stay is a good measurement tool that may partially provide some information on social distancing [15]. Google uses the location services that are available in Android devices. Depending on the products users use and settings they choose, they may provide Google with different types of location information. Location is provided from real-time signals, like the IP address or device location, and also past activity on Google sites and services. This information is collected and calculated by the start time and end time. For instance, if a user stayed 3 h in a shopping center, then Google considers that period the length of stay. This length of stay is compared to previous values recorded before the pandemic. The previous length of stay is known as the baseline. A lower value for the length of stay indicates a tighter lockdown.

Google reports included the following locations, retail and recreation, grocery and pharmacy, parks, transit stations, workplaces, and residential. All these values were given a percentage of comparison with a baseline of zero. A minus value represents a very low length of stay, which can be interpreted as a lockdown policy. A higher value with a minus sign represents a stricter lockdown. This is true for all locations except residential since residents will stay the longest time at home. Thus, data show a decrease in six attributes and an increase in the residential attribute simultaneously.

In addition to the Google specified locations, we added schools as one more location, since they were missing in the Google reports. Google uses mobile location services to prepare these reports, which are not available for school children. The school's data were abstracted from the UNESCO website and given a value of −30 on the closure date and onward. Zero value was assigned to the school opening dates [16]. The dataset was cleaned and prepared, as shown in Table 2. Data were analyzed based on seven independent variables, as mentioned before, and one dependent class. The dependent class was chosen from the R0 mean values of the interval >Q3. This interval was chosen since it measures the latest periods. The dates chosen were between 15 February 2020 and 4 November 2020, for 98 countries. Table 2 lists sample data for Australia with the dependent and independent attributes.

The aim of this dataset is to study the efficiency of social distancing and lockdown measures at any time. The data were classified using various models of Decision Table, Random Forest (RF), and K-Nearest Neighbor (KNN). RF is a classifier that produces multiple decision trees, using a randomly selected subset of training samples and variables [17], while KNN assumes that similar things exist in close proximity and are near to each other [18]. In the initial stage of analytics, the dataset was classified for all countries. Two models were used in these analytics, KNN and Random Forest. The data split was conducted by the test mode of 3-fold cross-validation for all datasets. KNN was used for the combination of all independent attribute classifications, while Random Forest was used for individual attribute classifications. Figure 1 shows the correlation coefficient average and the mean absolute error (MAE) for all countries. The MAE measures the average magnitude of the errors in a set of forecasts without considering their direction.

Figure 1 shows a low correlation coefficient and a high MAE for each individual factor, for instance, Retail and Recreation do not have a direct impact on the on Q3, since the correlation coefficient (CC) is around 0.35. It was concluded from Figure 1 that individual factors do not have a direct impact on lockdown efficiency. All independent attributes were found to have between 0.32 and 0.38 correlation values with R0 mean values. The correlation value was higher when combining all independent attributes since it reached up to 0.68. Moreover, the MAE was found to be the least value when a combination of all attributes was chosen.

### 4.1. Grouping Lockdown Efficiency by Countries

In the second stage of data analytics, we compared several countries. We chose a few countries with very high-level lockdowns with curfews, such as Italy, Jordan, and Indonesia. Then, we compared these countries to countries with fewer restrictions on lockdown procedures. The results showed a large variance in correlation values between countries. Figure 2 illustrates the correlation coefficients and MAE for 13 countries. We only considered the models that provided the highest value of CC and the lowest value of MAE. In all experiments, we ensured that the Relative Absolute Error does not exceed 50% as much as possible. The resulting values indicated that some countries had forced the lockdown policy while it is ineffective enough, such as Italy and India. Based on the resulting output from analyzing various countries, lockdown efficiency can be categorized into four main R0 groups, high (A), medium (B), low (C), and very low (D):R0 Group A: High lockdown efficiency is given by (CC ≥ 80%)R0 Group B: Medium lockdown efficiency is given by (60% ≤ CC < 80%).R0 Group C: Low lockdown efficiency is given by (40% ≤ CC < 60%).R0 Group D: Very low lockdown efficiency is given by (−40% < CC < 40%).

As mentioned earlier, there are many factors that may affect the infection and mortality rates. The value of R0 is only one of these factors. Therefore, Figure 2 and Figure 3 show the lockdown policy impact on R0 only for two different periods. Figure 2 shows values up till April, while Figure 3 shows values up till July. Figure 2 shows two countries from group D, Germany and Spain. This may indicate that Germany and Spain are not gaining any productive results out of the lockdown procedures. Figure 3 shows a dramatic increase in group rank. Here, Germany jumped from group D to group B, while Spain jumped from group D to group C. The South Korean government claimed that they did not apply any lockdown procedures. However, referring to the Google report, the lockdown was clearly applied to retail shops and transportation. It seems that the lockdown in South Korea was voluntary, hence, the R0 value dropped from R0 = 0.6 in February to R0 = 0.02 in April. In Figure 3, South Korea moved from group C to group A. As shown in Figure 2, India is in group C, though it started a very strict lockdown. Therefore, this may improve the lockdown efficiency in the coming weeks. Figure 3 illustrates the second result of the Indian lockdown by moving from group C to group B. In Brazil’s case, Figure 2 and Figure 3 show that Brazil remained in group B, which does not reflect the real status of the infection rate. This may indicate that Brazil has other factors apart from the Google parameters affecting lockdown efficiency. The Google report shows high lockdown values, yet their infection rate was skyrocketing. Moreover, the R0 value was low since April. This may give some indications about other factors that governments need to consider. Late procedures could be a major reason behind the large outbreak. The United States has a similar case to Brazil since late lockdown procedures intensively contributed to the current outbreak.

The high value of MAE may indicate that these seven factors are not the only factors. There might be other unknown factors that have a negative impact on lockdown efficiency. This measurement does not give a direct indication of exponential increase or decrease in the infection numbers. Instead, it only indicates lockdown efficiency.

### 4.2. Methodology Drawbacks

This model aimed to give an approximate value for R0 and to provide some factors that may have direct impacts on R0. However, this model is unable to resolve all the factors that may increase infection and mortality rates during the outbreak; instead, it presents the R0 value and the changes that occur by applying lockdown procedures. The lockdown procedures should reduce the R0 value but not the number of infections because R0 is only one out of many factors that may affect the infection rate. Many factors may increase the infection rate such as R0, late government action, number of ports in land and water, airport access policies, a country’s homogeneity or heterogeneity, lifestyle, and others.

Moreover, Google reports have limitations in measuring the length of stay due to some technical issues. For instance, many users may disable the location features in their mobile devices, or might not have mobiles supported by Android. Some countries may not have reliable access to the internet. Other users may have less mobility such as elderly people living in geriatric healthcare centers. In addition, some countries that were most affected by the pandemic did not have Google access such as Iran and China. Other countries did not apply any lockdown policies. All these reasons may reduce Google report accuracy.

## 5. Conclusions

A statistical model is defined in this paper. These calculations may support government plans and decision making. Since this pandemic may last a year or more, then there should be a well-structured plan to resume usual life activities with a high level of caution. The statistical model was based on Google reports on social distancing for measuring lockdown efficiency. The aim was to find the correlation coefficient for seven independent attributes with one class related to R0 value abstracted from the third part (Q3) of the interquartile range. The model was applied to 13 different countries. The model showed a significant correlation between the tight lockdown and the number of infections. The lockdown efficiency was categorized into four main R0 groups, high (A), medium (B), low (C), and very low (D).

## Figures and Tables

**Figure 1 ijerph-17-05574-f001:**
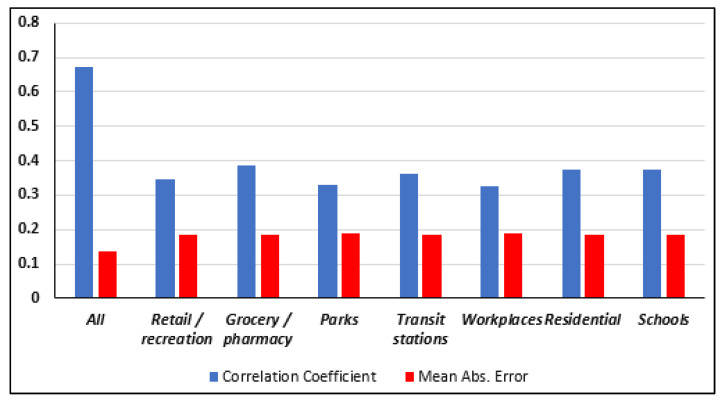
Correlation coefficients for all countries.

**Figure 2 ijerph-17-05574-f002:**
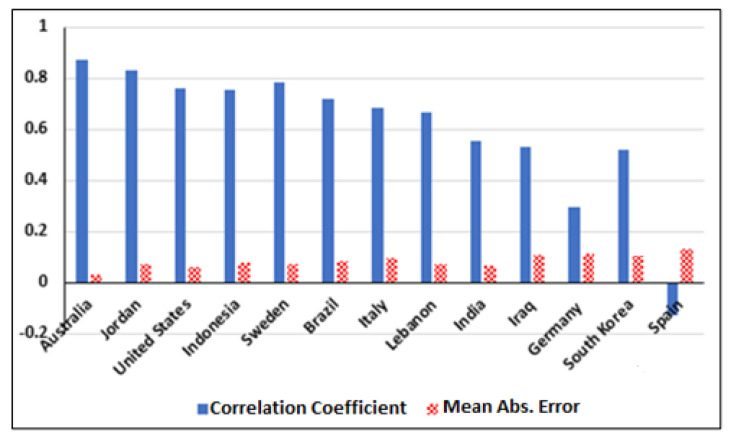
Correlation coefficients for 13 countries during the period between 15 February 2020 and 11 April 2020.

**Figure 3 ijerph-17-05574-f003:**
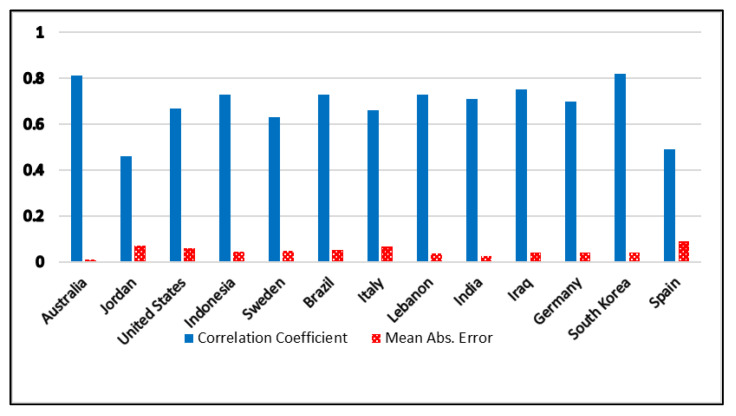
Correlation coefficients for 13 countries during the period between 15 February 2020 and 05 July 2020.

**Table 1 ijerph-17-05574-t001:** Comparison between COVID-19, SARS, MERS, and EVD.

Description	COVID-19 ^1^	SARS ^2^	MERS ^3^	EVD ^4^
Incubation period	2–14 days	2–10 days	2–14 days	2–21 days
Contagious during incubation	Yes	Yes	Yes	No
Vaccine release year	N/A	N/A	N/A	Dec. 2019
Infected body part	Blood vessels	Respiratory system	Respiratory system	All muscles
Latest outbreak	Dec. 2019	Nov. 2002–July 2003	April 2012	Dec. 2013–June 2016
			July 2015	
Winter/summer impact	Likely	Likely	Not likely	Not likely
Main transmission	Respiratory droplet secretions	Respiratory droplet secretions	Respiratory droplet secretions	Body fluids
Shape and size in nm	Spherical 80–120 nm	Spherical 80–90 nm	Spherical 90–125 nm	Filament
				14,000 × 80 nm
Temperature impact	Likely	Likely	Likely	Likely
Mortality rate	2.65%	14–15%	34%	Up to 90%
Pronounced R-naught	2–2.5	3.1–4.2	<1	1.5–1.9

1. COVID-19: Coronavirus Disease 2019; 2. SARS: Severe Acute Respiratory Syndrome; 3. MERS: Middle East respiratory syndrome; 4. EVD: Ebola Virus Disease.

**Table 2 ijerph-17-05574-t002:** Dataset abstracted from UNESCO [16] and Google reports [15].

Country	Date	>Q3 ^1^	Retail and Recreation	Grocery/Pharmacy	Parks	Transit Stations	Workplaces	Residential	Schools
Australia	3/20/20	0.22	−12	20	−12	−22	0	6	0
Australia	3/21/20	0.25	−17	11	−9	−28	−4	7	0
Australia	3/22/20	0.26	−18	10	−11	−34	−12	6	0
Australia	3/23/20	0.25	−17	17	−27	−36	−6	8	0
Australia	3/24/20	0.25	−30	4	−30	−45	−19	12	−30
Australia	3/25/20	0.25	−31	2	−33	−50	−23	14	−30
Australia	3/26/20	0.25	−34	2	−23	−53	−27	16	−30
Australia	3/27/20	0.24	−35	2	−30	−53	−26	17	−30

^1^ Q3: The third interquartile.

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
