# Peer review of "Measurement Method for Evaluating the Lockdown Policies during the COVID-19 Pandemic"

_ijerph, 2020, doi:10.3390/ijerph17155574_

Round 1
Reviewer 1 Report
REVIEW - "Measurement Method for Evaluating the Lockdown Policies During the COVID-19 Pandemic"
This manuscript is very interesting and very dare at the same time. The main problem of write something dare is build something with so many elements that need improvement (it is a slow and hard work). There are some suggestions that could help the authors with the task (anyway these suggestions will not exclude the considerable risk level of this research).
MATERIAL AND METHODS: There is a problem at this second sector of the manuscript: this part does not explain about materials or any method that were used at the research. Maybe chapter 2 could be a final part of the introduction and chapter 3 would be the beginning of material and methods sector. Nevertheless, chapter 3 needs to be adapted carefully because this sector is being a methodological and result/discussion chapter at the same time.
Lines 92-93: "The measurements model in this paper is the measurement of the lockdown procedures and finding out the most appropriate regulation that governments should take". - This is a strong affirmation of the manuscript. One thing is try to find a way to quantify a lockdown proceeding, however, find an appropriate regulation to governments actions is another issue. Be careful.
Lines 96-101: These lines have the same text of lines 92-96.
Lines 111-112: "A period of four days is defined as a short time to re-review the public health policy". - Four days would be a short time to review some public health actions for a better spatial protection level against this disease, but it is not be possible to review a health policy in my opinion.
Lines 167-174: It is relevant a short explanation about what is the methods K-Nearest Neighbour, Random Forest and Mean Absolute Error (it would be very didactical into a common table). Moreover, it is important explain better GOOGLE statistical method "length of stay" because this technique is the spinal foundament of your method proposition. Notwithstanding,
"length of stay" could not answer properly about lockdown efficiency in my opinion, however, it is an interesting tool for a possible analyzing process.
Sub-Chapter 3.2 - That is compulsory to show again the period of collecting data at sector 3.2 (15th Feb. to 11th Apr) and explain how was the current COVID pandemic panorama in each all these selected countries within this collection data period. I am writing this because your results certainly would bring polemical interpretations of Figure 2. I am thinking about some questions that the readers could criticize you:
1) Brazil did not developed an efficient lockdown process until nowadays as well. In fact, this country has presenting many strategical failures about lockdown processes since the beginning of COVID spread. How could the authors explain the Brazilian statistical result with this cited panorama?;
2) South Korea is showing one of the best examples of country prevention against COVID. If the authors are writing that this country is a successful case (lines 206-207), how South Korea has achieving this good condition?
3) Germany has showing one of the best examples of health sector action if we compare numbers of death people of its neighbour countries. If Germany did not developed an efficient lockdown, how this country could achieved better results than France and UK?
4) How the United States correlation coefficient could explain the terrible American situation?
These four questions maybe could change the pure statistical aspect of the research and decrease partially the risk of some possible gap into your model and within the final results indeed.
Author Response
Thank you for your valuable feedback. Our responses in the attachment.

Reviewer 2 Report
Dear authors,
Thank you for submitting this manuscript and your efforts to contribute to build evidence on how to evaluate lockdown strategies following the Covid 19 pandemic.
I have some suggestions to further improve the manuscript;
Introduction: Please declare your aim for this paper in the end of this section. I also think that you need to define or describe your definition of a community lockdown, and potential positive and negative health effects.
I think that the comparison between COVID-19, SARS, MER and EVD is interesting and important, but not part of the materials and methods, but the background.
In the section 3. Measurement Methods, I´m not really clear on what you focus on? Is it the previously used methods, or the methodology you used for this study? Please try to focus the section on the methodology used in this study.
As an overall comment, I think that you need to help the reader to understand the basics behind models to evaluate effects from lockdown on the spreading of a decease, by declaring what variables you include in the model, and what the outcome value is (R0).
Much of your model is based on Google reports. However, I miss the discussion on the limitations of Google in areas or countries not having access to reliable internet, or the value for populations that are not on internet or moving around in the society, such as elderly people living in geriatric health care centers. They are in many countries the most affected, and not supported by the lockdown strategy. Neither, spreading within the health care services is not included in the model, and that is also a limitation.
Otherwise, I think the paper is clear, and I hope to read a revised version shortly!
Author Response

(The authors gave the same response as above.)

Round 2
Reviewer 1 Report
The article untitled "Measurement Method for Evaluating the Lockdown Policies During the COVID-19 Pandemic" has improved its quality. Moreover, it is better developed in its results, discussion sector, and methodology indeed (it is considering some limits of the methodology, which is a normal proceeding at all).
The research is relevant and fine to be published. Thank you.